# Non-Invasive Delivery of Negatively Charged Nanobodies by Anodal Iontophoresis: When Electroosmosis Dominates Electromigration

**DOI:** 10.3390/pharmaceutics16040539

**Published:** 2024-04-13

**Authors:** Phedra Firdaws Sahraoui, Oscar Vadas, Yogeshvar N. Kalia

**Affiliations:** 1School of Pharmaceutical Sciences, University of Geneva, CMU-1 Rue Michel Servet, 1211 Geneva, Switzerland; phedra.sahraoui@unige.ch; 2Institute of Pharmaceutical Sciences of Western Switzerland, University of Geneva, CMU-1 Rue Michel Servet, 1211 Geneva, Switzerland; 3Department of Microbiology and Molecular Medicine, Faculty of Medicine, University of Geneva, CMU-1 Rue Michel Servet, 1211 Geneva, Switzerland; oscar.vadas@unige.ch

**Keywords:** nanobodies, anti-EGFR 7D12 nanobody, protein expression and purification, iontophoresis, electroosmosis, skin topical and transdermal delivery

## Abstract

Iontophoresis enables the non-invasive transdermal delivery of moderately-sized proteins and the needle-free cutaneous delivery of antibodies. However, simple descriptors of protein characteristics cannot accurately predict the feasibility of iontophoretic transport. This study investigated the cathodal and anodal iontophoretic transport of the negatively charged M7D12H nanobody and a series of negatively charged variants with single amino acid substitutions. Surprisingly, M7D12H and its variants were only delivered transdermally by anodal iontophoresis. In contrast, transdermal permeation after cathodal iontophoresis and passive diffusion was <LOQ. The anodal iontophoretic delivery of these negatively charged proteins was achieved because electroosmosis was the dominant electrotransport mechanism. Cutaneous deposition after the anodal iontophoresis of M7D12H_WT_ (wild type), and the R54E and K65E variants, was statistically superior to that after cathodal iontophoresis (6.07 ± 2.11, 9.22 ± 0.80, and 14.45 ± 3.45 μg/cm^2^, versus 1.12 ± 0.30, 0.72 ± 0.27, and 0.46 ± 0.07 µg/cm^2^, respectively). This was not the case for S102E, where cutaneous deposition after anodal and cathodal iontophoresis was 11.89 ± 0.87 and 8.33 ± 2.62 µg/cm^2^, respectively; thus, a single amino acid substitution appeared to be sufficient to impact the iontophoretic transport of a 17.5 kDa protein. Visualization studies using immunofluorescent labeling showed that skin transport of M7D12H_WT_ was achieved via the intercellular and follicular routes.

## 1. Introduction

The high potency, selectivity, and specificity of peptides and recombinant proteins make them excellent therapeutic agents. Nevertheless, their physicochemical attributes, namely their hydrophilicity, high molecular weight, poor stability in the gastrointestinal tract and, in some cases, short biological half-lives, limit their administration to the parenteral route (most commonly by subcutaneous or intravenous injection and by the intramuscular route for some sustained-release dosage forms). Despite the development of auto-injectors and fine-gauge needles, there can be issues of poor patient compliance, e.g., due to needle phobia, and the risk of unnecessary side effects, especially if systemic administration is used for localized medical conditions [1]. Furthermore, the benefit–risk profile can affect the target-patient population—the disease must be sufficiently severe or widespread to justify the systemic administration of a given biologic. Methods for local delivery to the site of disease would alter the benefit–risk profile and enable more patients to benefit from these potent therapeutics [2,3,4].

Iontophoresis is a non-invasive physical enhancement method for controlled topical and transdermal delivery. It uses a mild electric current (usually ≤0.5 mA/cm^2^) to facilitate the penetration of water-soluble ionizable molecules across the lipophilic stratum corneum [5] into and through the skin [6]. In addition to the avoidance of the gastrointestinal tract and hepatic first-pass metabolism, it also enables the tight control of drug input kinetics by the modulation of the current profile, customization through the adjustment of the current density, and is capable of mimicking physiological secretion patterns [7] that might otherwise only be achieved by intravenous infusion. In addition, this technique benefits from its ease of application, safety, patient convenience (self-administration capability), and the possibility of miniaturization [1,3].

Iontophoretic transport is governed by two principal mechanisms: electromigration (EM), which consists of directional ion flow under the influence of an electric field, and electroosmosis (EO), defined as the convective solvent flow due to the application of an electric field across a charged membrane. Given that the skin is negatively charged under physiological conditions (skin pI is between 4 and 4.5), electroosmotic solvent flow is in the anode-to-cathode direction. EO favors the anodal transport of positively charged or neutral compounds but hinders the cathodal electromigration of anions, indicating the cation-permselectivity of the skin. The principal driving forces depending on the charged species and iontophoresis type are depicted in Figure 1.

Peptides and proteins are usually charged at physiological pH; however, they have much higher molecular weights than the hydrophilic charged molecules typically delivered by iontophoresis. Nevertheless, several reports have demonstrated the iontophoretic delivery of moderately sized biologically active proteins such as cytochrome c (Cyt c) [8], ribonuclease A and T1 (RNase A and T1) [9,10], and human basic fibroblast growth factor (hbFGF) [11], into and through the skin. Surprisingly, electromigration was found to be the predominant mechanism for the transport of these proteins, although electroosmosis had been assumed to become the primary electrotransport mechanism over electromigration for “high” molecular-weight compounds [6,12,13], where “high” is assumed to be ≥1 kDa. More recently, cetuximab (CTX), a 152 kDa monoclonal antibody, was successfully delivered into the skin using this non-invasive technique [14], but in this case, electroosmosis was indeed the predominant mechanism.

It would be interesting to identify the inflection point, when electroosmosis would become the predominant electrotransport mechanism. However, although the proof-of-principle has been established, simple molecular descriptors such as mass, charge, pI, and electrophoretic mobility (which depends on the mass-to-charge ratio of the molecule), are poor predictors of the feasibility of the transdermal iontophoretic transport of proteins and the identification of the electrotransport mechanism. Previous work showed that although lysozyme had a higher electrophoretic mobility than Cyt c and RNase A, it had by far the poorest iontophoretic delivery [15,16]. This contradicted the hypothesis that increased electrophoretic mobility automatically favored iontophoretic delivery through increased EM. Indeed, interactions with the skin transport pathways are closely related to the conformation and surface-exposed residues of the protein, which exert a non-negligible effect on iontophoretic transport behavior [15,16]. This effect could already be seen with a series of tripeptides generated by changing the sequence of the amino acids at positions 6, 7, and 8 in nafarelin (D-Nal-Leu-Arg), i.e. where the tripeptides contained the same amino acids (hence, the same molecular weight and “log P”) but in a different order, displayed different iontophoretic delivery [17]. 

To investigate the effects of amino acid sequence and the spatial distribution of protein physicochemical properties on electrotransport, it is necessary to have a robust protein scaffold that can be modified and tested. Nearly 30 years ago, Hamers-Casterman et al. first discovered that camelids produce antibodies devoid of light chains and lacking the first constant C_H_1 domain, which reduces their size from 150 kDa to about 95 kDa [18]. The antigen-binding region of this special class of antibodies consists of a single variable domain referred to as V_H_H or Nanobodies^®^ (Nbs), a name given by Albynx (a subsidiary of Sanofi) because of their nanoscale size (4 nm × 2.5 nm × 3 nm). These single-domain antibody fragments are the smallest natural antigen-binding modules derived from naturally occurring heavy-chain-only antibodies (HcAb) present in the serum of Camelidae (i.e., *Lama guanaco*, *Camelus bactrianus*, *Lama Glama*, *Camelus dromedarius*, *Lama vicugna*, and *Lama alpaca*) [19]. These nanobodies exhibit remarkable properties compared to conventional antibodies, such as a small size (usually between 12–15 kDa, almost 10 times smaller than a classical IgG [20]), extreme stability, strong antigen-binding affinity (in the low nanomolar to picomolar range), deep tissue penetration, ease of engineering and production in simple expression systems such as *Escherichia coli* and *Pichia pastoris* [21,22], water solubility, the capacity to refold after denaturation while retaining their binding capacity, and amenability to be engineered into multivalent and multi-specific formats to increase the half-life, for instance, or to produce bispecific antibodies [23,24,25]. These favorable characteristics make nanobodies attractive diagnostic tools, probes, and potential next-generation biotherapeutics [26,27,28,29].

The anti-EGFR 7D12 nanobody (MW 13.4 kDa), which contains 124 amino acid residues folded into a globular structure (Figure 2), was selected as an interesting candidate to further explore the iontophoretic transport of biomacromolecules because of (i) its remarkable “nanobody” properties compared to conventional small proteins and/or antibodies, (ii) its structural similarities to RNase A, RNase T1, and Cyt c, and (iii) as a competitive EGFR inhibitor; like cetuximab, it could have a potentially exploitable therapeutic effect in dermatological conditions, such as non-melanoma skin cancers, atopic dermatitis, and psoriasis [30].

The specific objectives of the present study were the following: (i) to design a robust bacterial-expression system to express and purify the double-tagged (Myc- and His-tag) wild-type 7D12 nanobody (M7D12H_WT_) and a series of mono-substituted variants, R54E, K65E, and S102E, in the amounts necessary for the transport studies, (ii) to study the anodal and cathodal iontophoretic delivery of M7D12H_WT_ and its monosubstituted variants, (iii) to identify the iontophoretic mechanism that governed electrotransport of these negatively charged proteins, and (iv) to visualize iontophoretic pathways in the skin of M7D12H_WT_.

## 2. Materials and Methods

### 2.1. Materials

Sodium chloride (NaCl), 4-(2-hydroxyethyl)-1-piperazineethanesulfonic acid (HEPES), tris(hydroxymethyl)aminomethane (TRIS), imidazole, ammonium persulfate (APS), tetramethylethylenediamine (TEMED), acrylamide 4K, sodium dodecyl sulfate (SDS), dithiothreitol (DTT), bacteriological grade agar, chloramphenicol, glycerol, and Bovine serum albumin (BSA) were purchased from Applichem GmbH (Darmstadt, Germany). Trifluoroacetic acid (extra pure 99%) and hydrochloric acid (HCl) were supplied by Acros Organics (Geel, Belgium). Phenylmethylsulfonyl fluoride (PMSF) was purchased from Fluka (Buchs, Switzerland). Ampicillin sodium salt, DNase, Dulbecco’s phosphate-buffered saline (DPBS), 3,3′,5,5′-tetramethylbenzidine (TMB), silver chloride (AgCl), silver wire, SealPlate^®^colorTab™, and mouse monoclonal anti-polyHistidine-Peroxidase antibody (#A7058-1VL) were ordered from Sigma-Aldrich (Steinheim, Germany). Isopropyl β-d-1-thiogalactopyranoside (IPTG) was provided by Fluorochem (Hadfield, UK). PageRuler™ prestained protein ladder 26616, and unstained protein ladder 26614 were ordered from ThermoFisher Scientific, Life technologies (Plan-les-Ouates, Switzerland). Mouse anti-His antibody was purchased from Qiagen AG (Hombrechtikon, Switzerland) and the mouse anti-Myc antibody (9E10) was obtained from ThermoFisher Scientific (Zug, Switzerland). Terrific-Broth (TB), lysogeny broth (LB), 2xYT broth, and dialysis membrane Spectra/Por^®^ 3—MWCO 3.5 kD, 18 mm were purchased from CarlRoth (Karlsruhe, Germany). 0.22 µm/0.45 µm PVDF filters, centrifugal filters (MWCO 10kDa cut-off, Amicon^®^ Ultra-15), and NovaBlue competent cells were provided by Merck Millipore (Darmstadt, Germany). BL21(DE3), TOP10 chemically competent *E. coli*, and UltraPure™ agarose were obtained from Invitrogen (Carlsbad, CA, USA). NucleoSpin^®^plasmid and NucleoSpin^®^ Gel and PCR clean-up kits were purchased from Macherey-Nagel (Düren, Germany). The ELISA 96-well half-area plates were purchased at Greiner bio-one (Frickenhause, Germany). Tween 20 was ordered from Applichem Axon Lab AG (Baden-Dättwil, Switzerland). Chicken anti-Myc tag antibody (#ab9109) was obtained from Abcam (Cambridge, UK), and Alexa Fluor™ 594 goat anti-mouse from Life Technologies (Eugene, OR, USA). Tygon^®^ LMT 55 tubing (3.17 mm ID, 4.87 mm OD, 0.9 mm wall thickness) used to prepare salt-bridge assemblies was purchased from Saint-Gobain (Courbevoie, France). UltraPure™ Agarose was obtained from Invitrogen (Carlsbad, CA, USA). The OCT embedding matrix for frozen sections was procured from Biosystems (Muttenz, Switzerland). All aqueous solutions were prepared using Ultra-pure water (Millipore Milli-Q Gard 1 Purification Pack resistivity > 18 MΩ·cm; Zug, Switzerland). All other chemicals and solvents were at least of analytical grade.

### 2.2. M7D12H Plasmid Design

The sequence of a 7D12 nanobody was reverse translated (Emboss backtranseq) with the codon usage of *E.coli* K12. A “Myc-tag” sequence EQKLISEEDL was added to the N-terminus and a “His10-tag” was added to the C-terminal end, separated from the nanobody sequence by a GSSGS flexible linker (see Figure 3). This construct is henceforth referred to as M7D12H_WT_. The M7D12H_WT_ synthetic DNA sequence was ordered from GenScript Biotech (Rijswijk The Netherlands) and cloned into a pET22b (+) expression vector between *NcoI* and *HindIII*. Codon optimization was performed on the nanobody sequence.

### 2.3. Site-Directed Mutagenesis

Site-directed mutagenesis (SDM) was used to mutate specific amino acids on the M7D12H_WT_ DNA sequence (a detailed protocol is provided in the Appendix A). Three different mono-substituted variants were generated and referred to as follows (see Figure 4):R54E: Arginine (R) at position 54 substituted by glutamic acid (E);K65E: Lysine (K) at position 65 replaced by glutamic acid (E);S102E: Serine (S) at position 102 substituted by glutamic acid (E).

Figure 5 shows the 3D structure, as well as the electrostatic potential on the protein surface and the hydrophobicity/hydrophilicity, respectively.

### 2.4. Nanobody Expression and Purification

The M7D12H_WT_ nanobody and its single-substituted mutants were expressed, purified, and characterized in-house. Full details are provided in the Appendix A.

### 2.5. Skin Source

Porcine ears were obtained from a local abattoir (CARRE; Rolle, Switzerland). The skin was first removed from the external side of the ear before being excised with an air-dermatome (Zimmer, Winterthur, Switzerland) to obtain samples with a thickness of ~700 µm. Hairs were removed using clippers and the excised skin samples were then punched out (PERKIN-ELMER, Überlingen, Germany) in circular discs of 22 mm, wrapped in Parafilm™, and stored at −20 °C for a maximum period of 3 months.

### 2.6. Nanobody Quantification by Enzyme-Linked Immunosorbent Assay (ELISA)

Nanobodies were quantified using an enzyme-linked immunosorbent assay (ELISA). A Greiner™ 96-well half-area plate was coated with 50 µL of a chicken anti-Myc tag antibody (2 µg/mL in PBS) and was covered with a seal plate film and incubated overnight in the fridge (4 °C). The plate was then washed once with 175 µL of wash buffer (PBST) (0.1% Tween 20 in PBS). After that, non-specific binding sites were blocked with 125 µL of blocking buffer (BSA 3% (*m*/*v*) in PBST), and the plate was covered with the seal plate film and incubated at 37 °C for 1 h. A triplicated wash step was then performed and 50 µL of samples or standard solution were subsequently introduced and incubated at RT for 2 h. The wash step was repeated three times before adding 50 µL/well of a mouse anti-His HRP-linked antibody solution to detect the nanobodies. The plate was then covered and incubated at RT for 1 h. After a final triplicated wash step, 50 µL TMB substrate was introduced into each well, and the plate was incubated at RT in the dark for 10 min. Finally, 50 µL/well of 1N HCl solution was added to stop the enzymatic reaction, and the plate was read at 450 nm using the CLARIOstar plate reader (BMG labtech, Ortenberg, Germany). A standard curve was constructed over a concentration range of 5 ng/mL to 5000 ng/mL with the nanobody in the corresponding matrix, and the fitting was performed using GraphPad Prism 9.4.1 software (Appendix A). The five-parameter logistic (5PL) function was used for the regression, with an R^2^ superior to 0.99. The ELISA quantitation method was validated according to ICH guidelines on the bioanalytical method [31] (complete details are provided in the Appendix A). The limit of detection (LOD) and the limit of quantification (LOQ) in the different matrices (Wash buffer (WB), extraction matrix (EMTX), and permeation matrix (PMTX)) are summarized in Table 1.

### 2.7. Protein Stability in the Presence of Skin and Current

To test the stability of M7D12H_WT_ and its variants in contact with the skin, 1 mg/mL of each protein solution was placed in Eppendorf tubes containing skin samples cut into small pieces and maintained at 32 °C. Samples were collected at t = 0 and after 8 h of incubation (t = 8 h) to undergo SDS-PAGE analysis. Proteins in PBST were used as the control. Protein stability in the presence of the current was tested with Franz diffusion cells. Briefly, 3.5 kDa molecular weight cut-off (MWCO) dialysis membranes were cut to size and clamped between the donor and the receptor compartments. The Franz cells were then placed in a water bath and subjected to a current density of 0.5 mA/cm^2^ for 8 h using Ag/AgCl electrodes and salt bridges. The proteins were sampled before (t = 0) and after current application for 8 h for characterization by SDS PAGE (additional information regarding the SDS-PAGE setup can be found in the Appendix A).

### 2.8. Nanobody Delivery Studies

#### 2.8.1. Iontophoresis Setup

Franz diffusion cells (area 0.64 cm^2^; 5 mL volume of the receiver compartment) were used for the transport studies. The dermatomed skin samples were clamped between the donor and the receptor compartments after thawing the skin in NaCl 0.9% for 15 min. The Franz cells were then placed in a water bath equilibrated at 32 °C and 0.5 mL of protein solution (~5 mg/mL (0.3 mM) in buffer (25 mM Hepes, 133 mM NaCl, pH 7.4) and were added in the donor compartment. PBST was added to the receptor compartment. For **cathodal** iontophoresis, the AgCl cathode (−) was connected to the donor compartment via a salt-bridge assembly (3% agarose in 0.9% NaCl), and the Ag anode (+) was placed in the receiver compartment. The Ag/AgCl electrodes were connected to a power supply (Kepco^®^ APH 1000 M, Flushing, NY, USA), and a constant current density of 0.5 mA/cm^2^ was applied. For **anodal** iontophoresis, the electrodes were inverted. The experiment was conducted for 8 h, after which the PBST was collected from the receiver compartment for subsequent transdermal permeation analysis. The diffusion cells were then dismantled, and the residual donor solution was removed from the skin surface by washing thoroughly with PBST and wiping with a cotton disc. The skin area in contact with the protein solution was punched out to obtain a disc with a surface area of 0.503 cm^2^. The skin was subsequently cut into small pieces, and the protein deposited in the skin was extracted overnight with PBST under agitation. The samples were then centrifuged at 10,000 rpm for 15 min and diluted, if necessary, before being assayed by the validated ELISA method quantitation. The passive diffusion of the protein across the skin in the absence of an electrical current was studied as a control.

#### 2.8.2. Using Acetaminophen as a Marker of Electroosmotic Solvent Flow

To monitor the electroosmotic solvent flow, acetaminophen (ACM) was iontophoresed (anodal) in the presence or absence of the proteins (M7D12H_WT_ and its variants) using the same setup described above (Section 2.8.1). In total, 1 mL of ACM solution (15 mM in PBST) was placed in the donor compartment of the Franz cells in the presence or absence (control) of the proteins (0.3 mM). Iontophoresis at 0.5 mA/cm^2^ was performed for 8 h (experiments were performed in triplicate). A passive delivery (i.e., without a current) of ACM was also realized. The amount of ACM permeated across the skin was quantified by UHPLC-UV, using an Acquity^®^ UPLC^®^ H-Class system coupled to a photodiode array (PDA) UV Detector (Waters; Milford, MA, USA). Isocratic separation was performed using a Waters XBridge^®^ BEH C18 column (100 × 2.1 mm, 2.5 μm) maintained at 30 °C. The mobile phase consisted of MeOH:water (20:80, *v*/*v*) + TFA 0.1%. The flow rate and the injection volume were 0.25 mL/min and 2 µL, respectively. All solvents were degassed prior to use. ACM was detected by its UV absorbance at 243 nm and eluted at 1.83 min. The limit of detection (LOD) and limit of quantification (LOQ) were 0.28 µg/mL and 0.85 μg/mL, respectively.

#### 2.8.3. Estimation of EM and EO Contributions

The experimental protein flux, $J_{exp}$, is calculated from the concentrations in the receiver compartment determined by ELISA. These concentrations will depend on the contributions of electromigration and electroosmosis, *J_EM_* and *J_EO_*, respectively, which can act in unison to reinforce iontophoretic transport or in opposition to oppose it, depending on the charge of the protein (P), the formulation composition, the pH-dependent degree of ionization of the fixed negative charges in the skin, which will influence the magnitude of the convective solvent flow, and the electrode polarity (anodal or cathodal iontophoresis) (Figure 1) [6,8,9,11,13,15]. Although iontophoretic transport can be opposed by either EM or by EO under different scenarios, $J_{exp}$ will obviously always be ≥0 since it is not possible to increase the concentration of the permeant in the donor compartment.

The electroosmotic contribution, $J_{EO}$, is expressed as the product of the volume flow, $V_{W\;}$, created by the applied potential gradient and the concentration of the protein, $C_{protein}$. The former is calculated from the ratio of the flux, $J_{ACM}$, and the donor concentration, $C_{ACM}$, of acetaminophen and is expressed as a volume flow per unit area per unit time. This supposes that the convective transport of the protein is not subject to size- or structure-dependent effects that might hinder its transport as compared to that of the much smaller ACM.

Thus, assuming negligible passive permeability, then, $J_{exp}$ can be represented using Equations (1)–(3) below. The equations have been annotated to indicate the direction of transport, anode-to-cathode (A→C) or cathode-to-anode (C→A). The sign convention used is that a “positive” term indicates a contribution and directionality that favors transport from the donor to the receiver compartment (as mentioned above, in all cases, $J_{exp}$ ≥ 0); thus, for Equations (2) and (3), when the negative term is equal to or greater than the positive term, there will be no protein detected in the receiver compartment.

Anodal iontophoresis for a positively charged protein P (+):(1)$$Anodal_{P\left(+\right)}:J_{exp}\left(\geq 0\right)=J_{EM}^{A\;\to C}+J_{EO}^{A\;\to C}$$Cathodal iontophoresis for a negatively charged protein P (−):(2)$$Cathodal_{P\left(-\right)}:J_{exp}\left(\geq 0\right)=J_{EM}^{C\;\to A}-J_{EO}^{A\;\to C}$$Anodal iontophoresis for a negatively charged protein P (−):(3)$$Anodal_{P\left(-\right)}:J_{exp}\left(\geq 0\right)=J_{EO}^{A\;\to C}-J_{EM}^{C\;\to A}$$
where $J_{EO}^{A\;\to C}=V_{W\;}\times C_{protein}$ and $V_{W}=J_{ACM}/C_{ACM}$.

### 2.9. Immunohistochemistry

#### 2.9.1. Microscope Sample Preparation

Upon completion of the iontophoresis experiment, the skin surface was cleaned as described previously, placed in a cryomold, and embedded in OCT to form frozen blocks after snap-freezing. Longitudinal slices of either 10 or 20 μm thickness were obtained using the cryotome (Thermo Scientific CryoStar™ NX70; Reinach, Switzerland) and were placed on microscope slides (SUPERFROST^®^ PLUS, Braunschweig, Germany). These OCT sections were fixed in a mixture of methanol and acetone (1:1) at −20 °C for 5 min. The slides were then washed once at RT in 0.05% Tween 20 in PBS for 3 min and in 0.1 M Tris in DPBS (pH 8) for 3 min before being blocked for at least 40–60 min in 4% normal goat serum (NGS) in PBS in a humidified chamber. After the blocking, the primary antibody (mouse anti-His antibody (Qiagen, #34660)) prepared in 4% NGS in PBS was applied to the sections, and they were covered with parafilm and incubated at RT for 1 h in a humidified chamber. After a duplicated wash step of 3 min using 0.05% Tween 20 in PBS, the secondary antibody (goat anti-mouse Alexa Fluor ™ 594 (Invitrogen, #A11032)) prepared in 4% NGS in PBS was applied to the sections and they were covered with parafilm and incubated at RT for 1 h in the same conditions as the primary antibody. Cell nuclei were then stained with DAPI for 5 min. After a final washing step in 0.1 M Tris in DPBS (pH 8) at RT, the slides were embedded using a EUKITT mounting medium and dried overnight in the dark after covering them with a coverslip.

#### 2.9.2. Microscope Images

Image acquisition was realized in the Bioimaging Core Facility of the Faculty of Medicine, University of Geneva. The visualization of the skin sections was performed using an Axio Imager Microscope (Axioscan.Z1, Carl Zeiss, Germany) using a 10× objective (Plan Apochromat 10×/0.45 M27). The excitation and emission wavelengths used for DAPI were 353 nm and 465 nm, respectively. For Alexa Fluor 594, the excitation and emission wavelengths were 590 nm and 618 nm, respectively. Images were all acquired and analyzed using the same exposure time (500 ms for Alexa Fluor 594 and 50.01 ms for DAPI), objective, and display-settings range (contrast, brightness, and color adjustment) to provide an accurate comparison of the signals. They were analyzed and processed using Zen Blue 3.1 and ImageJ 1.52n software.

### 2.10. Statistical Analysis

Skin delivery data were expressed as mean ± SD. Outliers were determined using the Grubbs test and were discarded. The results were evaluated statistically using GraphPad Prism 9.3.1. Groups were compared using analysis of variance (one-way or two-way ANOVA). Dunnett’s or Tukey’s multiple comparisons tests were used as follow-up tests. 

## 3. Results and Discussion

### 3.1. M7D12H Plasmid Design

The pET-22b (+) vector was used to express the protein in the bacterial periplasm. Indeed, this plasmid carries an N-terminal *pelB* signal peptide which, when attached to a protein, directs the latter to the periplasmic space where this short leader sequence of 22 amino acids is then removed by a signal peptidase [32,33]. The Myc-tag, attached to the target protein’s N-terminus, serves for ELISA quantification, while the His-tag at the C-terminus acts both as a purification tag (IMAC) and for ELISA quantification.

The purpose of using tags was to establish a universal quantification method that was applicable to both the M7D12H_WT_ and its variants, regardless of any protein mutations. This was necessary because the antibody pair used in the ELISA specifically recognized the tags and not the recombinant protein itself.

### 3.2. Nanobody Expression and Purification

The purification protocol was stable and highly efficient. Table 2 shows the approximate quantities of protein routinely obtained after purification.

### 3.3. Protein Stability in the Presence of Skin and Current

Neither exposure to porcine skin nor the application of current for 8 h resulted in any detrimental effects on the structural integrity of the nanobodies, as confirmed by the analysis of SDS-PAGE gels. More details can be found in the Appendix A (Appendix A).

### 3.4. Investigation of Nanobody Iontophoretic Transport

#### 3.4.1. Cathodal Iontophoresis

Given the results obtained with RNase T1 [10], where cathodal iontophoresis resulted in the electromigration-driven transport of RNase T1 into and across the skin, it was decided to begin the transport experiments with an investigation of the cathodal iontophoretic delivery of the negatively charged M7D12H_WT_ and its variants. However, and in contrast to RNase T1, the transdermal permeation of both M7D12H_WT_ and its variants after cathodal iontophoresis (and after passive delivery) was below the LOQ. Although extraction studies showed that there was skin deposition of M7D12H_WT_ and its variants, there were no statistically significant differences between the amounts of each protein retained in the skin after cathodal iontophoresis or passive delivery (Figure 6a). These quantities were very small and similar for the wild-type protein, R54E, and K65E (1.12 ± 0.30, 0.72 ± 0.27, and 0.46 ± 0.07 µg/cm^2^, respectively). Interestingly, the post-cathodal iontophoresis skin deposition of the S102E variant (8.33 ± 2.62 µg/cm^2^) was much greater than that of its congeners, and the difference was statistically significant (*p* < 0.05) (Figure 6a). This suggested that the modification of serine at position 102 by glutamic acid may have facilitated its electromigration. However, given the increased passive delivery and the variability, there was no statistically significant superiority of the post-iontophoretic deposition of the S102E variant over that seen in the absence of current application.

Overall, the results suggested that cathodal transport by electromigration was insufficient to overcome electroosmosis in the opposite direction (anode-to-cathode), perhaps due to the lack of sufficient negative charge at pH 7.4, which limited electrophoretic mobility (Table 3). More evidence of the impact and predominance of electroosmosis came from the observation that protein could be extracted from the salt bridges connecting the donor compartment to the cathodal compartment: thus, M7D12H_WT_ and the variants were being driven from the donor compartment into the salt bridges by the magnitude of the anode-to-cathode electroosmotic solvent flow (Figure 6b).

Based on these observations, it was decided to investigate the feasibility of using anodal iontophoresis, and hence, electroosmosis, to deliver the negatively charged M7D12H_WT_ and its variants.

#### 3.4.2. Anodal Iontophoresis

The cumulative permeation of M7D12H_WT_ and its variants as a function of time during 8 h of anodal iontophoresis at a current density of 0.5 mA/cm^2^ across porcine skin and the total amounts permeated after 8 h are shown in Figure 7a,b, respectively. Surprisingly, the negatively charged M7D12H_WT_ nanobody and its variants were all detected in the receiver compartment; thus, they were successfully delivered transdermally using anodal iontophoresis despite the opposing effect of electromigration in the cathode-to-anode direction. The extraction experiments revealed that there was a statistically significant difference between proteins deposited in the skin after the completion of the anodal iontophoresis and passive delivery (Figure 7c and Table 4), further confirming the improvement in the delivery of these proteins to the skin using this non-invasive technique. The quantities recovered were 6.07 ± 2.11, 11.89 ± 0.87, 9.22 ± 0.80, and 14.45 ± 3.45 µg/cm^2^ for M7D12H_WT_, S102E, R54E, and K65E, respectively. The total delivery, i.e., the sum of the amounts extracted and permeated, is presented in Figure 7d. The data are summarized in Table 4. Interestingly, the cumulative permeation of M7D12H_WT_ and S102E proteins at t = 8 h was very similar and was calculated to be 11.39 ± 8.30 and 11.63 ± 5.67 µg/cm^2^, respectively. Similarly, the quantities permeated for R54E and K65E were 2.94 ± 0.99 µg/cm^2^ and 4.18 ± 2.04 µg/cm^2^, respectively (Table 4).

These results pointed to the key role of electroosmosis and how it governed electrotransport since electromigration was in the opposite, i.e., cathode-to-anode, direction.

Figure 8 compares the skin deposition of the M7D12H_WT_ nanobody and its variants after anodal and cathodal iontophoresis. The superiority of anodal iontophoretic transport due to electroosmosis is clear for three out of the four proteins tested. Interestingly, since the deposition of the S102E variant following cathodal iontophoresis was much greater than the other proteins, the difference between its skin deposition after anodal and cathodal iontophoresis was not statistically significant.

#### 3.4.3. Acetaminophen Co-Iontophoresis

Electroosmosis (EO) depends on the convective solvent flow from the anode-to-the-cathode direction upon the application of the electric field across a charged membrane, such as skin. Indeed, the acidic nature of the skin (the pH ranges between 4.2 and 6.1) leads to a pH gradient going from the stratum corneum to the deepest layers of the viable epidermis where the pH is fixed at 7.4 [34,35,36]. Since the skin’s isoelectric point (pI) ranges between 4 and 4.5, this means that it is negatively charged at a physiological pH (7.4) (pH > pI) [37]. As a result, EO generates a volume flow in the direction that favors the movement of counterions to neutralize the skin’s net charge; hence, at physiological pH, this is in the anode-to-cathode direction and facilitates cation transport. This mechanism promotes the electrotransport of neutral molecules from the anode and reinforces the anodal delivery of cations but opposes the cathodal electromigration of anions [12,38].

Acetaminophen (ACM) is added to the donor compartment of an anodal iontophoretic system since, as a neutral molecule, it is transported only by electroosmosis [39,40]. As such, ACM was used here as a marker to estimate the electroosmotic contribution to nanobody transport. In addition, the co-iontophoresis of ACM with, usually, a cationic protein reveals whether the latter binds to negative charges of the skin, thereby inhibiting the electroosmotic flow, and so can be used to determine the inhibition factor.

Figure 9 shows the cumulative acetaminophen permeation as a function of time for 8 h (Figure 9a) and the cumulative amounts delivered after 8 h (Figure 9b). The data are presented in Table 5. The inhibition factor is represented by the ratio between cumulative ACM permeation in the absence and the presence of the protein of interest (Q_ACM Prot (-)/_Q_ACM Prot (+)_). The inhibition factor (IF) of M7D12H_WT_, S102E, R54E, and K65E variants was calculated to be 1.47, 1.96, 1.65, and 1.00, respectively.

The S102E mutant showed the highest inhibition factor (IF = 1.96) compared to the other three proteins, indicating a stronger interaction with the skin’s negative charges. When added to the donor compartment, S102E led to a significant decrease (*p* < 0.05) in ACM permeation, demonstrating EO inhibition, similarly observed with the R54E mutant (Figure 9b). However, the presence of the M7D12H_WT_ protein or K65E did not significantly impact ACM permeation (*p* > 0.05), as reflected by their IF values (IF = 1.47 and 1.00, respectively). These initial findings highlight how a single amino acid substitution can tip the balance of the IF of M7D12H_WT_ from 1.47 to 1.00 or ~2 (double).

Interestingly, S102E and K65E exhibited comparable depositions (11.89 ± 0.87 and 14.45 ± 3.45 µg/cm^2^, respectively) despite K65E not inhibiting EO (IF = 1.00). Positively charged proteins, such as lysozymes, can neutralize the fixed negative charges, thus facilitating skin deposition and impacting EO [16]. Anionic hydrophilic proteins (negative GRAVY values, Table 3) like these nanobodies will clearly interact differently and may possess fewer possible binding sites and might also be susceptible to electrostatic repulsion. This suggested that the inhibition of solvent flow cannot be attributed to the negation of the fixed negative charges but also to other phenomena due to protein accumulation in the membrane that might hinder transport. This hypothesis was also advanced to explain the results seen with RNase T1 [10].

The results confirmed that the anodal iontophoretic delivery of a 17.5 kDa negatively charged protein was driven by the electroosmotic solvent flow (Table 6), which was previously reported to be the electrotransport mechanism for the iontophoresis of anionic carboxy inulin across hairless mouse skin [41]. Indeed, unlike RNase T1 [10], electromigration, in the cathode-to-anode direction, was not sufficient to overcome the opposing electroosmotic solvent flow, which explains why the experimental protein flux (J_exp. protein_) was low. Therefore, for anionic proteins with physicochemical properties more closely aligned to these nanobodies, it would be more appropriate to perform anodal iontophoresis since EO appears to be the main electrotransport mechanism. The results further demonstrate that single amino acid substitution in a protein of 160 residues (MW~17.5 kDa) is sufficient to have an impact on the interaction with skin and on protein electrotransport.

### 3.5. Therapeutic Relevance of the Iontophoretic Delivery of 7D12

The epidermal growth receptor (EGFR or ErbB1) is most prevalent in the basal epidermis and the outer root sheath of the hair follicle. Its overexpression or mutation can lead to several malignancies, including basal and squamous cell carcinomas, head and neck carcinomas, non-small-cell lung cancer, and colorectal cancer, to name only a few [42,43,44]. Cetuximab, a recombinant IgG1 human–murine chimeric monoclonal antibody, binds to the extracellular domain and competitively inhibits the EGFR and has been used, in combination, to treat regionally advanced or metastatic squamous cell carcinomas of the head and colorectal cancer [45]. As mentioned above, it can also have an application in the targeted, localized treatment of dermatological conditions including atopic dermatitis and psoriasis [30].

The anti-EGFR 7D12 nanobody sterically blocks the ligand-binding region on domain III of the EGFR, in a similar manner to cetuximab, and it is also able to compete with the antibody for EGFR binding. This nanobody, therefore, has the potential to mimic the clinical efficacy of cetuximab in a more stable agent that is much less expensive to produce [46]. In addition, multivalent/multi-specific nanobody formats can be created to recognize different epitopes on the EGFR, thereby generating even more potent therapeutic agents. Along these lines, Roovers et al. [47] developed a biparatopic version of an anti-EGFR nanobody, combining the specificities of cetuximab and matuzumab (COoperative NANobody-1 –“*CONAN-1*”–), which inhibits tumor outgrowth in vivo with cetuximab-like potency, in athymic mouse models bearing subcutaneous A431 xenografts.

The dissociation constant (K_D_) of the 7D12 nanobody toward the epidermoid squamous carcinoma A431 cells was measured to be 25.7 nM [47]. The amounts extracted and permeated for the wild-type protein following anodal iontophoresis were calculated to be approximately equal to 6 and 11 µg/cm^2^, respectively, which corresponds to approximately 86 and 157 µg/mL (assuming a skin thickness of 700 µm), i.e., 5 and 9 µM (MW = 17,487.1 kDa), respectively. This is equivalent to ~192- and ~350-fold the K_D_. Without optimizing the delivery, the concentrations achieved already exceed those required for receptor blockade. Therefore, pharmacologically relevant concentrations of the nanobody could theoretically be achieved using anodal iontophoresis. These preliminary results on the feasibility of delivery are promising, but further studies need to be performed in relevant disease models to demonstrate pharmacological efficacy using more patient-friendly conditions.

### 3.6. Visualization of Penetration Pathways

The extraction data showed that significant quantities of M7D12H_WT_ were retained in the skin after anodal iontophoresis. It was therefore decided to study the nanobody distribution and penetration pathways in the skin by visualizing their transit through the membrane.

Figure 10 shows images of the skin after the M7D12H_WT_ delivery experiments after (a) cathodal or (b) anodal iontophoresis and (c) passive diffusion, and (d) the blank control (i.e., no M7D12H_WT_ protein). The images showed that M7D12H_WT_ was mainly localized in the epidermis, particularly in the stratum corneum, with high intensity after anodal iontophoresis compared with cathodal iontophoresis and passive delivery where the intensities were comparable and much lower. Moreover, the immunofluorescence signal was also detected in the dermis after anodal iontophoresis, as depicted in Figure 10b, showing the diffusion of the protein from the epidermis to the dermis, which is consistent with the results of the protein-permeation data (Table 4). Indeed, M7D12H_WT_ was delivered transdermally after 8 h of anodal iontophoresis, demonstrating the penetration of the protein in deeper tissue. Moreover, the Alexa Fluor™ 594 signal was clearly seen in the surrounding keratinocytes, suggesting an intercellular pathway. Indeed, it has been reported that iontophoretic current application on the skin surface leads to an activation of an intracellular signaling pathway that results in the opening of the intercellular junctions, thus contributing to the transport of substances, such as biomacromolecules across the skin barrier [1]: it was reported that this was due to lowering the amount of the gap-junction protein connexin 43 and by triggering the depolymerization of the F-actin linked to tight junctions [48]. The faint red signal observed in Figure 10d was due to the background fluorescence signal.

Skin appendageal structures include sweat glands, hair follicles, and sebaceous glands, and a significant fraction of iontophoretic transport is thought to occur through these structures, particularly in the hair follicles, as this is the path of least resistance to current flow. Although the appendages occupy a small fraction of the skin surface area, they can constitute a major penetration pathway for permeants with poor diffusion through bulk skin. Indeed, it has been reported that the transdermal permeation of biologics by iontophoresis occurs mainly through the transappendegeal pathways and also through the paracellular route by opening the intercellular junctions under the application of an electric field, creating a rapid-transport shunt [1,49]. Figure 11 clearly shows the immunofluorescence signal in the skin appendages, specifically in the hair follicles after iontophoretic or passive delivery. Indeed, the signal was detected in the inner and outer root sheaths of the hair follicles. The latter is in continuity with the basal layer of the epidermis and is generally rooted in the dermis, which explains the detection of the nanobody in this skin layer [50,51,52].

Taken together, these micrographs provide striking visual evidence that anodal iontophoresis enabled much deeper penetration of the M7D12H_WT_ nanobody through the skin layers than cathodal iontophoresis and passive delivery, which is consistent with the previous quantitative results. Furthermore, skin penetration occurred primarily via the appendageal and intercellular routes; a very interesting observation given the expression of the EGFR in the outer root sheath of the hair follicle.

## 4. Conclusions

The results presented in this study demonstrate the feasibility of the topical and transdermal delivery of nanobodies to porcine skin in a targeted and completely non-invasive manner. Delivery was significantly greater after anodal iontophoresis and was governed by electroosmosis. To the best of our knowledge, this is the first report demonstrating the successful non-invasive anodal iontophoresis by electroosmosis of a negatively charged high-MW protein across intact porcine skin, which is the best surrogate for human skin.

It was also concluded that a single amino acid substitution was sufficient to have an impact on the interaction with the skin and protein electrotransport. Furthermore, the amounts extracted and permeated across the skin after anodal iontophoresis may be sufficient to consider the non-invasive administration of therapeutic quantities of M7D12H_WT_. In addition, it was determined that the skin penetration pathways for the latter were primarily intercellular and follicular.

The tertiary and quaternary structure of proteins is complex and can have a highly variable impact on their iontophoretic transdermal delivery. Indeed, they can interact very differently with the cutaneous transport pathways, making it difficult to predict their transport. To gain further insight into the factors controlling protein electrotransport, it would be of great interest to conduct additional studies with a larger number of replicates and varying the number of mutations. These studies would enable the development of a multivariate regression model to predict protein electrotransport more accurately and identify suitable candidates for iontophoresis.

## Figures and Tables

**Figure 1 pharmaceutics-16-00539-f001:**
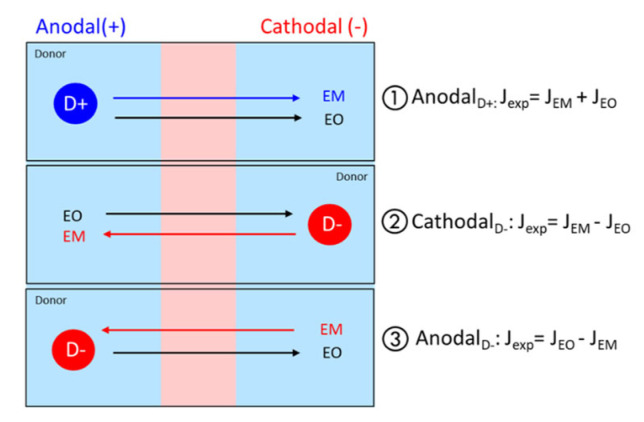
Iontophoretic transport is governed by the contributions of electromigration (EM) and electroosmosis (EO). Their magnitude and relative contribution is a function of the permeant (anion or cation) in the donor compartment, the isoelectric point of the skin (pI), and the type of iontophoresis (anodal or cathodal). They can act in conjunction or in opposition.

**Figure 2 pharmaceutics-16-00539-f002:**
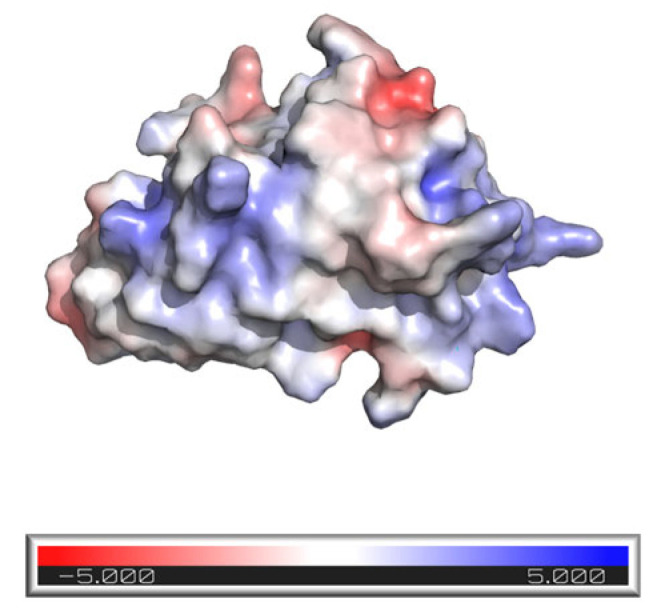
3D structure of the 7D12 nanobody displaying the electrostatic potential onto the molecular surface (Connolly surface). Blue and red colors represent regions of high positive and negative charge density, respectively. Software used: AlphaFold2.ipynb for protein structure prediction and Pymol for visualization.

**Figure 3 pharmaceutics-16-00539-f003:**
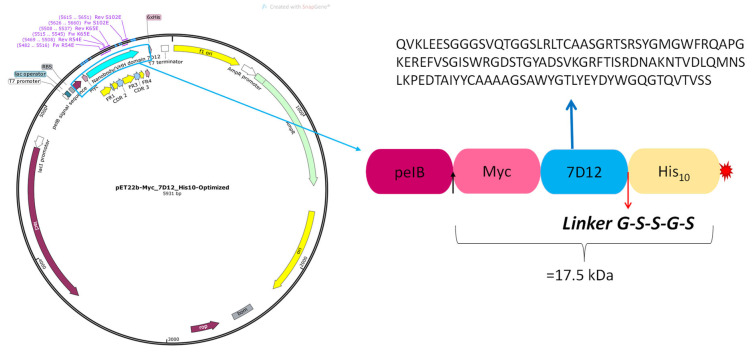
Plasmid map of M7D12H_WT_ and its corresponding sequence.

**Figure 4 pharmaceutics-16-00539-f004:**
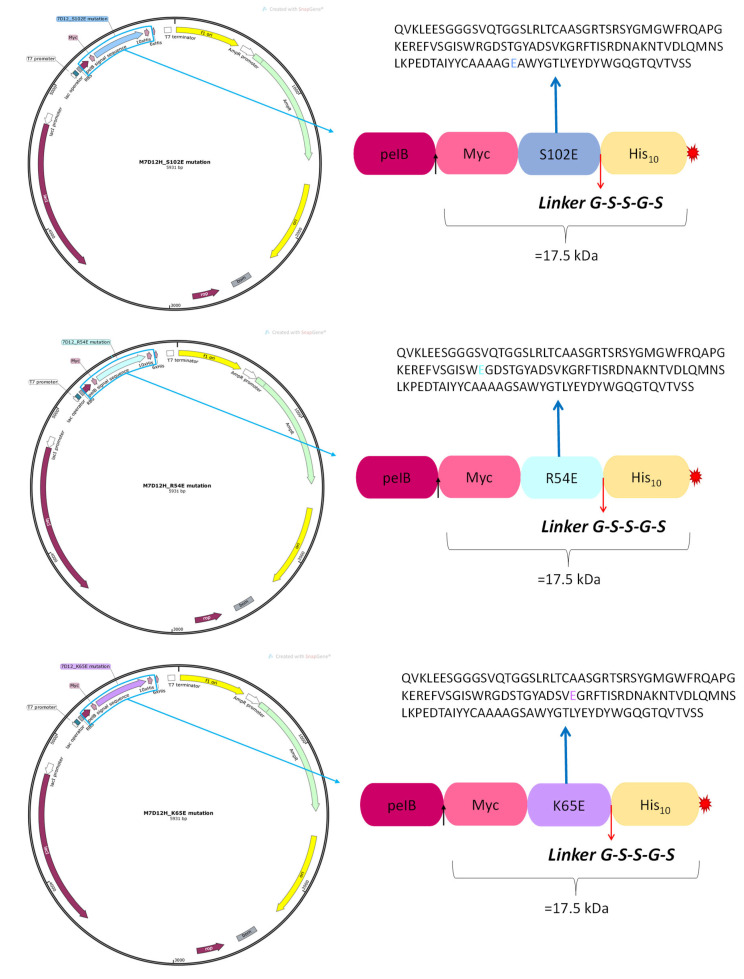
Plasmid maps of the mono-substituted variants and their corresponding sequences.

**Figure 5 pharmaceutics-16-00539-f005:**
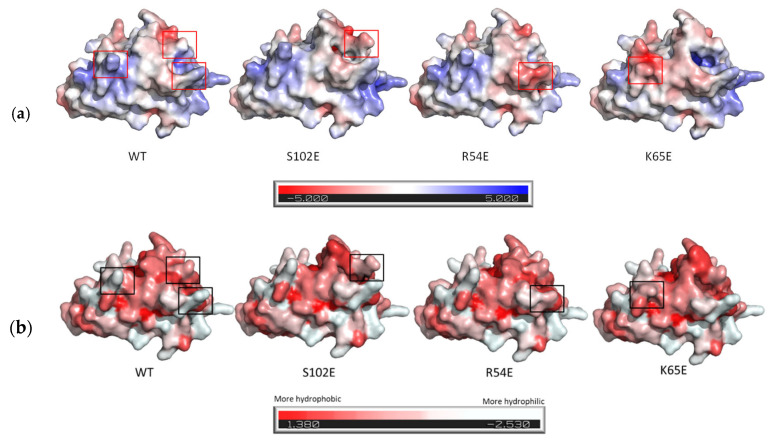
(**a**) 3D structure of the 7D12 nanobody and its variants displaying the electrostatic potential on the molecular surface (Connolly surface). Blue and red colors represent regions of high positive and negative charge density, respectively; (**b**) coloration of protein residues according to the Eisenberg hydrophobicity scale. Red and white colors represent regions of high hydrophobicity and high hydrophilicity, respectively. Software used: AlphaFold2.ipynb for protein structure prediction and Pymol for visualization.

**Figure 6 pharmaceutics-16-00539-f006:**
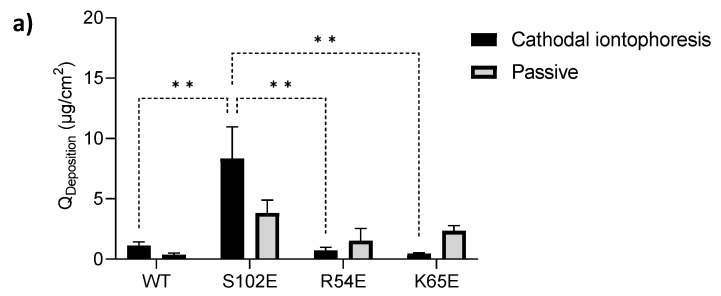
(**a**) Skin deposition of M7D12H_WT_ nanobody and its variants following 8 h cathodal iontophoresis at 0.5 mA/cm^2^ across porcine skin and a comparison between S102E deposition with the wild-type protein and R54E and K65E mutants. (**b**) Extraction from saline bridges after cathodal iontophoresis, (Mean ± SD; *n* = 3 for iontophoresis and *n* = 2 for passive control). ** indicates a *p* value ≤ 0.01.

**Figure 7 pharmaceutics-16-00539-f007:**
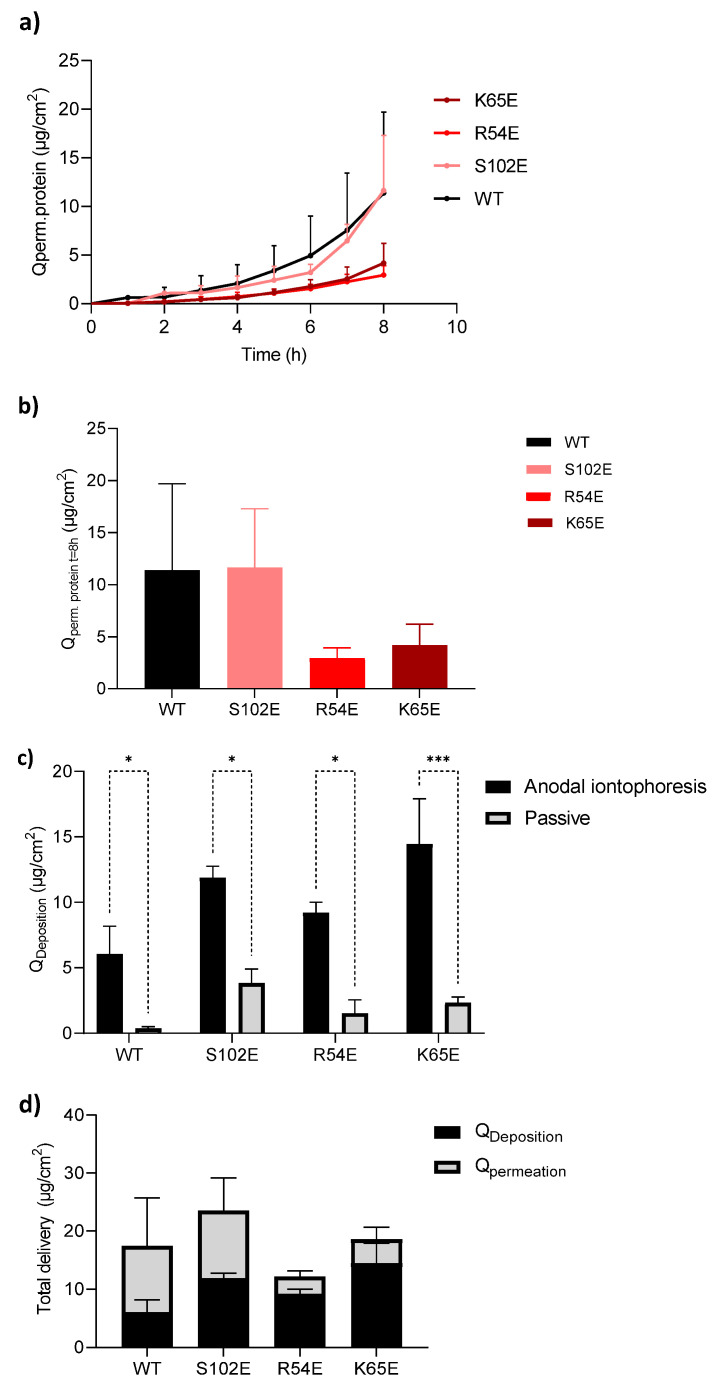
(**a**) Cumulative protein permeation (M7D12H_WT_ and its variants) as a function of time during 8 h of transdermal iontophoresis at 0.5 mA/cm^2^ across porcine skin; (**b**) total amounts permeated after 8 h of iontophoresis; (**c**) skin deposition; and (**d**) total delivery of M7D12H_WT_ and its variants after anodal iontophoresis. (Mean ± SD; *n* = 3 for iontophoresis and *n* = 2 for passive control). * indicates a *p* value ≤ 0.05, *** indicates a *p* value ≤ 0.001.

**Figure 8 pharmaceutics-16-00539-f008:**
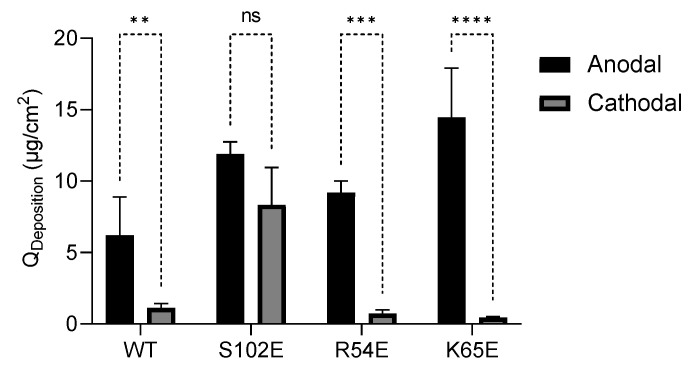
Comparison of anodal and cathodal iontophoretic delivery. (Mean ± SD; *n* = 3). ** indicates a *p* value ≤ 0.01, *** indicates a *p* value ≤ 0.001, and **** indicates a *p* value ≤ 0.0001.

**Figure 9 pharmaceutics-16-00539-f009:**
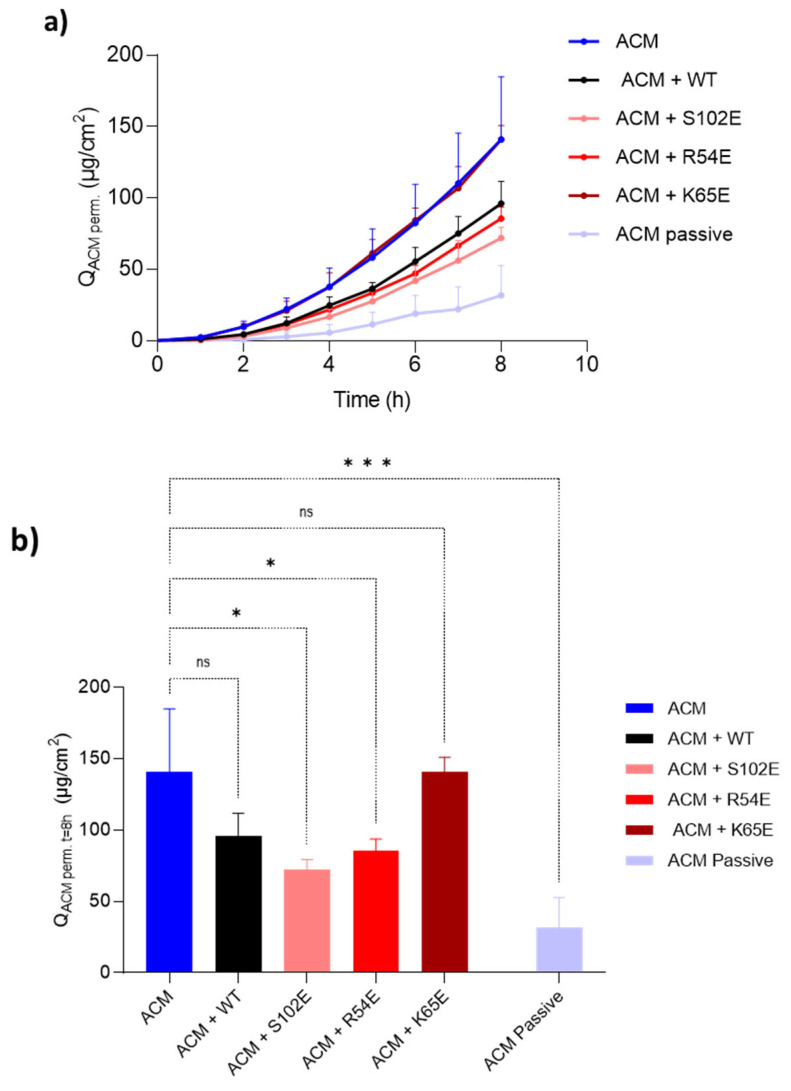
(**a**) Cumulative acetaminophen permeation as a function of time during 8 h of transdermal iontophoresis at 0.5 mA/cm^2^ across porcine skin in the presence and the absence of M7D12H_WT_ and its variants and (**b**) cumulative amounts delivered after 8 h of iontophoresis. (Means ± SD; *n* = 3 for iontophoresis and *n* = 2 for passive control). * indicates a *p* value ≤ 0.05, *** indicates a *p* value ≤ 0.001.

**Figure 10 pharmaceutics-16-00539-f010:**
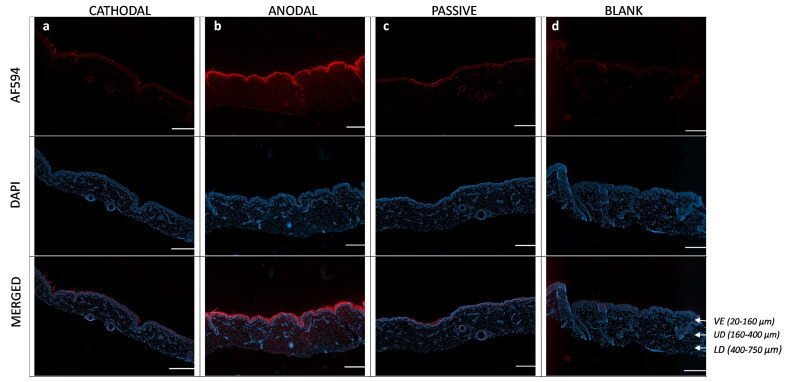
Immunofluorescence labeling studies of M7D12H_WT_ distribution across porcine skin (longitudinal sections) following either 8 h of constant current anodal or cathodal iontophoresis at 0.5 mA/cm^2^ ((**a**) and (**b**), respectively), passive diffusion for 8 h (**c**), and untreated porcine skin (control) (**d**). The samples were labeled with Alexa Fluor™ 594 goat anti-mouse IgG antibody, and cell nuclei were stained with DAPI. The images contain an overlay (superimposition) of M7D12H_WT_ (red) and nuclei (blue). They were taken with a widefield scanner Zeiss Axioscan.Z1 at 10× magnification and treated with ImageJ software. VE: viable epidermis; UD: upper dermis; LD: lower dermis. Scale bar = 500 µm.

**Figure 11 pharmaceutics-16-00539-f011:**
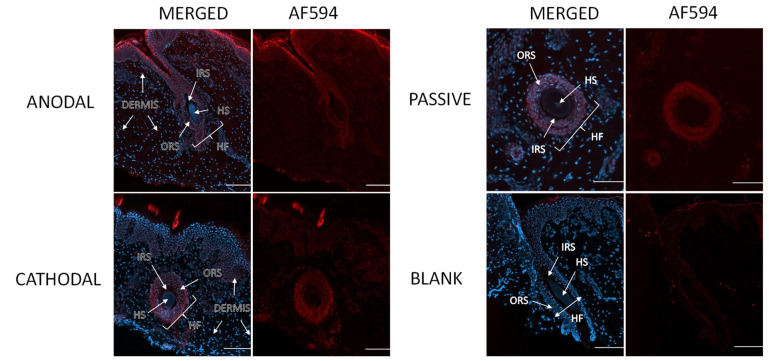
Localization of M7D12H_WT_ nanobody in hair follicles after anodal iontophoresis, cathodal iontophoresis, and passive delivery. Blank: porcine skin incubated with primary and secondary antibodies. HF: hair follicle; ORS: outer root sheath; IRS: inner root sheath; HS: hair shaft. Images were obtained with a widefield scanner Zeiss Axioscan.Z1 at 10× magnification and treated with ImageJ software. Scale bar 100 μm.

**Table 1 pharmaceutics-16-00539-t001:** Determination of LOD and LOQ in the different matrices.

	WB	EMTX	PMTX
LOQ (ng/mL)	5.18	5.14	5.07
LOD (ng/mL)	1.73	1.71	1.69

**Table 2 pharmaceutics-16-00539-t002:** Amount of protein obtained after purification for each variant per liter of culture.

	M7D12H_WT_	S102E	R54E	K65E
Approximate amount of protein per liter of culture (mg/L)	23–28	22	17	16
Mass of bacterial pellet (g)	7.9	7.9	6.1	9.8
Protein quantity (mg/g bacteria)	3.2	2.8	2.8	1.6

**Table 3 pharmaceutics-16-00539-t003:** Comparison of physicochemical properties of the M7D12H_WT_ and its variants. The GRAVY value is an indicator of the hydrophobicity/hydrophilicity of a protein.

	MW (Da)	pI (Isoelectric Point)	Z (Net Charge at pH 7.4)	GRAVY
M7D12H_WT_	17486.89	6.50	−3.35	−0.703
S102E	17528.93	6.33	−4.35	−0.720
R54E	17459.82	6.17	−5.35	−0.697
K65E	17487.84	6.17	−5.34	−0.701

**Table 4 pharmaceutics-16-00539-t004:** Total delivery of M7D12H_WT_ and its variants after anodal iontophoresis.

Anodal Iontophoresis		
	Q_Deposition_ (µg·cm^−2^)	Q_Permeation_ (µg·cm^−2^)
M7D12H_WT_	6.07 ± 2.11	11.39 ± 8.30
S102E	11.89 ± 0.87	11.63 ± 5.67
R54E	9.22 ± 0.80	2.94 ± 0.99
K65E	14.45 ± 3.45	4.18 ± 2.04

**Table 5 pharmaceutics-16-00539-t005:** Iontophoretic transport ACM in the presence and absence of M7D12H_WT_ and its variants and the corresponding inhibition factors (Mean ± SD; *n* = 3).

	Q_ACM perm. t = 8h_ Cumulative Permeation (µg·cm^−2^)	IF (Q_ACM Prot(−)_/Q_ACM Prot(+)_)	ACM Flux J_ACM 5–8 h_ (µg·cm^−2^·h^−1^)	V_w_ (J_ACM_/C_ACM donor_ ) (µL·cm^−2^·h^−1^)
**ACM**	140.96 ± 43.94	─	27.59 ± 8.00	12.17 ± 3.53
**ACM +** M7D12H_WT_	96.08 ± 15.55	1.47	19.85 ± 3.62	8.76 ± 1.60
**ACM + S102E**	71.93 ± 7.32	1.96	14.75 ± 0.84	6.50 ± 0.37
**ACM + R54E**	85.52 ± 8.05	1.65	17.52 ± 2.68	7.73 ± 1.18
**ACM + K65E**	140.83 ± 9.93	1.00	26.15 ± 1.95	11.53 ± 0.86

**Table 6 pharmaceutics-16-00539-t006:** Iontophoretic transport kinetics of M7D12H_WT_ and its variants and the relative contributions of electromigration and electroosmosis (Mean ± SD; n = 3).

	Protein Flux J_exp. protein 4–8 h_ (µg·cm^−2^·h^−1^)	J_EO protein cacluated_ (µg·cm^−2^·h^−1^)	J_EM protein estimated_ (µg·cm^−2^·h^−1^)
M7D12H_WT_	3.53 ± 1.81	44.12	40.59
S102E	4.20 ± 3.24	29.92	25.72
R54E	0.56 ± 0.25	38.94	38.38
K65E	0.85 ± 0.44	51.78	50.93

## Data Availability

The data presented in this study are available on request from the corresponding author.

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
