# Peer review of "Non-Invasive Delivery of Negatively Charged Nanobodies by Anodal Iontophoresis: When Electroosmosis Dominates Electromigration"

_pharmaceutics, 2024, doi:10.3390/pharmaceutics16040539_

Round 1

Reviewer 1 Report

Comments and Suggestions for Authors

The study introduces an innovative approach to the non-invasive transdermal delivery of nanobodies using iontophoresis, addressing a significant challenge in the field of drug delivery. A comprehensive investigation of both anode and cathode iontophoresis for the delivery of negatively charged M7D12H nanobody and their variants demonstrates a robust experimental design. This manuscript is very impressive, however, major revisions are needed before it can be published:

1.   Even with the same mechanism as drugs such as cetuximab and matuzumab compared in the manuscript, it is difficult to argue that it works with simple calculations, as claimed between the 553-563 lines, without confirming its validity in in vivo experiments. Expanding the therapeutic relevance of iontophoresis delivery of M7D12H nanobody, especially the context of skin cancer treatments, will strengthen the paper. Including preliminary in vivo efficacy data can highlight the clinical potential of this delivery method if possible.

2.   The results shown in Figures 7, 8, and 9 demonstrate the outcomes of an 8-hour iontophoresis treatment. However, this duration may be deemed impractical for actual clinical practice. It would be beneficial to obtain experimental findings that display statistically significant differences in a shorter timeframe.

3.   Although Figure 7a and Figure 7b show exactly the same results, there is room for confusion as if they were different results, so it seems necessary to express them as one.

4.   On page 17 of supplementary materials, figure s9b and figure s9c need to be changed to figure s8b and figure s8c.

Reviewer 2 Report

Comments and Suggestions for Authors

The paper investigated transdermal iontophoretic delivery of nanobodies.  The experimental designs are sound.  The results and discussion are thorough.  However, there are concerns that require justification/ clarification.  Below are the point-by-point comments.

Major concerns:

1.   The equations in Section 2.8.3 are not entirely correct.  The assumption of additivity of the fluxes (Jem and Jeo) will give a negative flux (a) when Jem is larger than Jeo for anodal iontophoresis or (b) when Jeo is larger than Jem for cathodal iontophoresis.  However, negative fluxes (net transport of drugs from the receptor to the donor) do not occur.  All the assumptions used in these equations should be clearly stated.

2.  The solvent permeability coefficients (Vw) for acetaminophen (ACM) and proteins are likely not the same.  For example, when there is transport hindrance due to the large molecular size of proteins across the skin (compared to the size of the pathways in the skin for iontophoretic transport), the Vw determined by ACM will be an overestimation of the solvent permeability coefficients for the proteins.  Hindered transport (size exclusion property) has been suggested for transdermal iontophoretic delivery in the literature.

3.  Why were the Q8h results (not the fluxes) used in the inhibition factor (IF) calculation in Table 5?  Using fluxes would provide somewhat different IF values because the use of Q8h was under the assumption that linear (steady state) permeation from the beginning of the experiment (t = 0 h), which was not the case.  Justifications are needed for the use of Q8h results and not the flux in this and other analyses in the paper.

4.  Were the Jem values in Table 6 calculated using the equations in Section 2.8.3?  From Comments #1 and 2 above, the analysis in Table 6 needs more discussion (and clarification). 

Minor concerns:

5.  The Introduction is long with figures.  There are 30 references used in the Introduction.  The figures are likely not necessarily (as these previous findings can be easily described in the text).  This is not a review paper.  The Introduction should be more concise. 

6.  It should be specified that the “poor predictors” are for iontophoretic transport across the skin in the 6th paragraph in the Introduction (line 90-98).

7.  Equation numbers are missing for the equations in Section 2.8.3.

8.  Although “WT” and “M7D12Hwt” are the same and can be used interchangeably, they can be confusing.  Using single term consistently in the paper is preferable. 

9.  The sentence “. . . EO generates a volume flow. . . that attempt to neutralize the skin’s negative charges” (line 477) is confusing.  This should be revised.

10.  The first sentence in the Conclusion (line 634) does not apply to passive transdermal delivery.  This should be specified. 

Round 2

Reviewer 2 Report

Comments and Suggestions for Authors

#1.  This reviewer disagrees with the authors on the explanation.  Negative flux from donor to receiver (i.e., net flux from receiver to donor) should not occur under the experimental conditions in this study because there is no protein in the receiver to provide the flux from the receiver (or transporting from the receiver to the donor).  The lowest flux (J exp) that can occur is zero.  It cannot be negative.  However, according to eq. 2, when J eo > J em, the experimental flux J exp is negative, implying that there can be protein flux from receiver to donor and an increase in concentration of the protein in the donor.  The same is for eq. 3 when J em > J eo.  This is not correct.  Using the boundary conditions of the concentration across the membrane and solving the differential equation of iontophoretic flux, the correct equation of J exp is a function of: (x)/(1 – exp (– (x))), where x is related to the electromigration term in the Nernst-Planck equation and the volume flow.  Such relationships can be found in iontophoresis literature, which do not give a negative flux when J eo > J em in cathodal iontophoresis (or J em > J eo in anodal iontophoresis) of a negatively charged protein.  This reviewer also disagrees with the explanation using “basis of reverse iontophoresis.”  Reverse iontophoresis occurs when there is an analyte in the receiver transporting to the donor (or adding the analyte in the receiver and measuring the analyte in the donor in the experiment).  If there is no analyte in the receiver, there is no flux.

This reviewer is not asking for a long explanation of the theory or modifying the equations in Section 2.8.3.  Adding the assumptions to specify the conditions so that J exp is not negative will be appropriate: e.g., dominant J em (or J em > J eo) for eq. 2 and dominant J eo (or J eo > J em) for eq. 3.  These conditions with dominant J from one condition over the other will allow J eo and J em to be additive (when the function (1-exp(-(x)) ≈ 1) and that J exp will not become negative (when x < 0).

#2.  This reviewer agrees with the explanation that “solvent flow is represented as the volume transported per unit area per unit time; hence, this should be independent of the permeant.”  However, the flux of the permeant transporting across skin under convective transport is size dependent.  Unlike convective transport in free solution, even though volume flow is constant, the volume flow effect is different on the flux of permeants of different molecular sizes due to hindered transport.  In this study, the volume flow flux measured by ACM is likely not the same as that of the protein.  An assumption in the analyses in the paper is that there was no hindered transport, and the effects of the convective transport on ACM and protein fluxes are the same.  Or, if the authors have evidence that the effects are the same, this should be provided.  Possible overestimation of the solvent permeability coefficients for the proteins by using the Vw determined by ACM should be discussed or at least mentioned. 

#4.  See #1 and #2 above.
